# Pyruvate Kinase M2 Links Metabolism and Epigenetics: A New Target for Liver Disease Treatment

**DOI:** 10.3390/biom15091327

**Published:** 2025-09-17

**Authors:** Xiaoya Zhang, Danmei Zhang, Jin Guo, Chunxia Shi, Zuojiong Gong

**Affiliations:** Department of Infectious Diseases, Renmin Hospital of Wuhan University, Wuhan 430060, China; 2023203020008@whu.edu.cn (X.Z.); 2021203020016@whu.edu.cn (D.Z.); 2020283020122@whu.edu.cn (J.G.); chunxiashi18@whu.edu.cn (C.S.)

**Keywords:** Pyruvate kinase M2, liver disease, post-translational modification, treatment

## Abstract

The aberrant activation of glycolysis plays a pivotal role in the progression of liver diseases. Pyruvate kinase M2 (PKM2), one of the rate-limiting enzymes of glycolysis, not only regulates cellular metabolism but also translocates to the nucleus in its dimeric form, acting as a co-factor to modulate gene transcription. To further explore the regulatory mechanisms of PKM2, this review outlines the effects of post-translational modifications on PKM2’s structure, activity, and localization, and discusses the integrative role of PKM2 in epigenetics and metabolism, providing a foundation for the development of PKM2 regulators. Due to PKM2’s distinct biochemical properties, targeting PKM2 with specific regulators may offer a promising therapeutic strategy for the treatment of liver diseases.

## 1. Introduction

Liver diseases, including simple fatty liver or non-alcoholic fatty liver disease (NAFLD), non-alcoholic steatohepatitis (NASH), liver failure, cirrhosis, and liver cancer, are experiencing a rising incidence [1]. Effective treatment options for certain liver diseases remain limited, likely due to a lack of in-depth research into the complex mechanisms underlying these conditions and the absence of well-defined drug targets.

The liver, as a central organ for the storage, synthesis, and metabolism of nutrients, plays a pivotal role in sensing nutrient input to maintain metabolic homeostasis [2]. However, during the progression of liver diseases, hepatocyte damage and inflammatory responses can lead to metabolic dysfunction, resulting in the aberrant activation of liver metabolism, including enhanced glycolysis [3]. In hepatocellular carcinoma (HCC), increased glycolysis supports HCC cell proliferation, invasion, migration, and the development of drug resistance [4]. Conversely, in other liver diseases, enhanced glycolysis promotes the activation of immune cells, contributing to ongoing liver damage and inflammation [5]. Therefore, investigating the role of aberrant glycolysis in liver diseases may provide effective strategies for combating these conditions.

Glycolysis is a major metabolic process that converts glucose into pyruvate, generating adenosine triphosphate (ATP) and nicotinamide adenine dinucleotide (NADH). Pyruvate kinase (PK), as one of the crucial enzymes in the glycolytic procedure, catalyzes the conversion of phosphoenolpyruvate into pyruvate in the final step of glycolysis, along with ATP generation [6]. In organisms, there exist four principal isoforms, including pyruvate kinase liver and red cell (PKL and PKR) encoded by the PKLR gene and pyruvate kinase muscle-isozymes M1(PKM1) and PKM2 encoded by the PKM gene [7]. Unlike other isoforms, PKM2 encompasses not just a highly active tetramer, promoting the flux of glycolytic metabolism towards oxidative phosphorylation (OXPHOS), but also less active monomers and dimers, diverting glycolytic metabolism from OXPHOS towards glycolysis [8,9]. Moreover, the PKM2 dimer can translocate to the nucleus, closely related to tissue repair, cell growth, and tumor development [10]. Indeed, due to its unique features, PKM2 has drawn much attention in disease study. Further investigations have uncovered that post-translational modifications of PKM2 play a key role in regulating its structure, function, and localization, creating the prerequisite for PKM2 in disease onset and progression (Figure 1).

In this review, we focus on the role of PKM2 in liver diseases, delve into the regulatory mechanisms of PKM2 and its role in epigenetics, and emphasize the therapeutic potential of PKM2 in liver diseases.

## 2. PKM2 and Liver Diseases

The expression levels of PK isoforms are tightly regulated in a tissue-specific manner. The PKL isoform is predominantly found in the liver, pancreas, and kidneys, while PKR is restricted to red blood cells. PKM1 is mainly expressed in muscle tissue, mature sperm, and the central nervous system, whereas PKM2 is highly expressed in the brain, kidneys, lungs, and spleen [11]. Although PKM2 expression is nearly undetectable in the healthy liver, it paradoxically represents the predominant PKM isoform in both normal and HCC liver tissues [12]. In HCC, PKM2 is enriched in extracellular vesicles [13] and correlates with poor prognosis [14]. Besides being overexpressed in HCC tissues, high PKM2 levels are also detected in precancerous cirrhotic tissues, closely associated with the high risk of HCC onset [15]. Elevated PKM2 levels are also observed in cirrhotic and NAFLD livers [16], indicating its potential as a diagnostic and prognostic biomarker for liver diseases (Figure 2).

### 2.1. The Nuclear Translocation of PKM2 Promotes Malignant Characteristics of HCC

Glycolysis supports HCC growth, metastasis, and immune evasion [3]. PKM2, a key glycolytic enzyme, exists in various conformational states. In its dimeric form, PKM2 exhibits low pyruvate kinase activity, thereby promoting the Warburg effect and enhancing HCC cell proliferation [17]. Importantly, dimeric PKM2 demonstrates a strong propensity for nuclear localization, where it functions as a transcriptional co-factor. Once translocated into the nucleus, PKM2 displays protein kinase activity—phosphorylating histone H3, suppressing histone deacetylation, and activating the transcription of downstream oncogenes—collectively contributing to tumor cell proliferation [18,19].

Post-translational modifications tightly regulate PKM2 localization and activity. Histone deacetylase 8 (HDAC8)-mediated deacetylation at K62 promotes PKM2 nuclear translocation and CCND1 transcription via β-catenin, while also regulating its enzymatic activity and glucose metabolism [20], while Guanosine triphosphate binding protein 4 (GTPBP4) -induced SUMOylation facilitates dimer formation and nuclear entry, activating STAT3 signaling and epithelial–mesenchymal transition in HCC [21]. Conversely, LncRNA PWRN1 inhibits tumor growth by maintaining PKM2 in its active tetrameric state, preventing nuclear translocation [22]. Therefore, nuclear localization of PKM2 plays an important role in tumor growth and metastasis.

Dimeric PKM2 also upregulates mitochondrial glutaminase 1 (GLS-1), coordinating the shift from glycolysis to glutaminolysis [23]. Both TNFα-stimulated gene 6 (TNFAIP6) [24] and methyltransferase-like protein 5 (METTL5) enhance PKM2 expression and activation through the c-Myc axis, promoting glycolytic reprogramming and contributing HCC progression [25]. This elevated glycolytic activity supports cancer cell proliferation, metastasis, and resistance to therapy. Once translocated into the nucleus, PKM2 enhances HIF-1α transactivation, further upregulating glycolytic genes and establishing a positive feedback loop that exacerbates drug resistance [26]. Notably, simvastatin has been shown to suppress PKM2-dependent glycolysis by targeting the HIF-1α/PPAR-γ/PKM2 axis, thereby inhibiting tumor cell proliferation, promoting apoptosis, and sensitizing HCC cells to sorafenib treatment [27].

PKM2 plays a critical role in shaping the immunosuppressive tumor microenvironment in HCC. Elevated PKM2 levels in HCC tissues are closely associated with increased infiltration of immune cells and heightened expression of immunosuppressive factors. Under high-glucose conditions, lactylation enhances PKM2 nuclear translocation, where it interacts with thioredoxin-1 (TRX1) to suppress chemerin expression, thereby promoting immune evasion and metastasis [28]. Furthermore, PKM2-driven phosphorylation of histone H3 is essential for the transcriptional activation of PD-L1, a key immune checkpoint protein that inhibits T cell proliferation and activity [18]. PKM2-dependent glycolysis also correlates with macrophage polarization, reduced antitumor immunity, and increased PD-L1 expression. Notably, the accumulation of PKM2^+^CD68^+^ macrophages in tumor tissues is linked to poor prognosis [14,15]. Paradoxically, high PKM2 expression may enhance sensitivity to immune checkpoint blockade, suggesting its potential as both a predictive marker and therapeutic target in immunotherapy. In addition, PKM2-enriched exosomes secreted by HCC cells promote M2-type macrophage polarization via STAT3 signaling, further contributing to immune suppression and tumor progression [13].

Clinical investigations have provided preliminary evidence linking PKM2 expression or circulating levels with disease severity and prognosis. In HCC, elevated PKM2 expression in tumor tissues has been correlated with aggressive clinicopathological features, including larger tumor size, higher tumor-node-metastasis (TNM) stage, and poorer overall survival [11,29]. Nevertheless, the clinical application of PKM2 as a biomarker faces several challenges. Its expression is not liver-specific, and elevated levels have also been documented in various malignancies and inflammatory conditions, potentially limiting its diagnostic specificity.

### 2.2. The Role of PKM2 in the Progression of Inflammation in Liver Diseases

Inflammation is a central pathological driver of many liver diseases. During hepatic steatosis, inflammatory responses accelerate disease progression from simple fatty liver to steatohepatitis [30]. In addition, excessive inflammation can trigger a cytokine storm, leading to extensive hepatocyte injury and death.

The pro-inflammatory role of PKM2 in liver disease is primarily mediated through metabolic reprogramming and activation of hepatic immune cells. In fibrotic liver tissues, Follistatin-like protein 1 (FSTL1) stabilizes PKM2 and promotes its nuclear translocation in macrophages, enhancing M1 polarization and the secretion of pro-inflammatory cytokines, thereby exacerbating fibrosis [31]. Additionally, PKM2-driven M1 macrophage polarization contributes to hepatocyte steatosis via paracrine signaling [32].

Hepatic stellate cell (HSC) activation, a key event in fibrosis, is associated with elevated PKM2 levels. Activated HSCs release PKM2-rich extracellular vesicles, which further stimulate macrophage activation and fibrosis [33]. Therapeutic agents such as TEPP-**46** and apigenin alleviate fibrosis by modulating PKM2 conformation and inhibiting its nuclear translocation [34,35]. Similarly, circ_0098181 exerts anti-fibrotic effects by directly binding PKM2, reducing its nuclear accumulation, and suppressing HSC activation and inflammation [36].

In hepatocytes, PKM2 activity is closely tied to lipid accumulation and inflammatory signaling [37]. In NAFLD, microRNA-122 downregulates PKM2, modulating inflammation and autophagy-related pathways [38]. Annexin A5 binds PKM2, inhibits its phosphorylation, promotes tetramer formation, and induces M2 macrophage polarization, thus attenuating steatohepatitis [39]. PKM2-rich extracellular vesicles promote macrophage activation and also play an important function in the development of alcoholic liver disease [40]. Beyond macrophages, PKM2 regulates Th17 cell metabolism, and its specific knockdown ameliorates liver inflammation and NAFLD progression [41]. In liver fibrosis, increased PKM2 expression has been observed in activated hepatic stellate cells and correlated with fibrotic stage in biopsy specimens [42], supporting its potential role as a marker of disease progression.

Acute liver failure (ALF) is a fatal syndrome driven by excessive inflammation. Nuclear-localized PKM2 interacts with STAT3, promoting its recruitment to the NLRP3 promoter region, thereby enhancing NLRP3 transcription and triggering the release of pro-inflammatory cytokines such as IL-1β and IL-18, ultimately contributing to ALF progression [43]. Bee venom peptides inhibit PKM2 activation, disrupt the Warburg effect, and reduce inflammatory injury in ALF [44]. Similarly, targeted depletion of PKM2 in macrophages significantly reduces inflammation and hepatocyte damage in acute liver injury models [45].

Recent evidence indicates that PKM2 is closely implicated in the pathogenesis of viral hepatitis and the progression of inflammation-driven liver diseases. Clinical metabolomics studies have demonstrated elevated serum PKM2 levels in patients with hepatitis B virus (HBV)-related cirrhosis and HCC, suggesting its potential as a noninvasive biomarker for virus-associated liver cancer [46]. Recent reviews further emphasize that PKM2 is upregulated within the inflammatory microenvironment of chronic liver diseases—including viral hepatitis, alcohol-related liver disease, and NAFLD—where it contributes to glycolytic reprogramming and immune regulation [1,47]. Moreover, blood proteomic profiling has identified PKM2 as one of the proteins elevated in cirrhotic patients at increased risk of developing HCC, many of whom had HBV-related cirrhosis [48]. Collectively, these findings support PKM2 as a promising biomarker and therapeutic target in hepatitis-related liver diseases, linking metabolic reprogramming with immune dysregulation and disease progression. Nevertheless, large-scale clinical validation is required before PKM2 can be reliably applied in the diagnosis or prognostic assessment of hepatitis.

Post-translational modifications serve as key regulators of PKM2’s diverse functions, while its nuclear translocation and subsequent regulation of gene expression represent critical mechanisms driving disease progression. Understanding its mechanistic regulation, including post-translational modifications and epigenetic roles, may provide critical insights for the development of targeted therapies in liver diseases.

## 3. The Post-Translational Modification of PKM2

Post-translational modifications (PTMs) are pivotal in regulating the diverse biological functions of PKM2, including its structure, enzymatic activity, subcellular localization, and roles in metabolism, tumor progression, and immune responses. These modifications—such as phosphorylation, acetylation, ubiquitination, SUMOylation, glycosylation, oxidation, succinylation, hydroxylation, and lactylation—form a complex regulatory network that precisely tunes PKM2 activity (Table 1).

### 3.1. Phosphorylation

Phosphorylation is one of the most critical PTMs of PKM2 [49]. For example, phosphorylation at Y105 disrupts FBP binding, supporting a glycolytic phenotype in cancer cells, while phosphorylation at S37 enhances nuclear translocation, influencing immune signaling [50,51]. On one hand, it serves as a co-activator of β-catenin in human glioblastoma cells, promoting tumor metastasis; on the other hand, in hepatocellular carcinoma, it promotes the phosphorylation of histone H3 at T11, inducing the expression of PD-L1, and inhibiting the proliferation and function of T cells, leading to immune suppression in the tumor microenvironment [15]. Conversely, phosphorylation at Thr365 by JNK1 enhances PKM2 activity and links glucose metabolism to apoptosis, suggesting potential therapeutic relevance [52].

### 3.2. Acetylation and Deacetylation

PKM2 undergoes acetylation at multiple lysine residues, including Lys433, Lys305, and Lys66—modifications that are reversible through the action of various deacetylases. Acetylation at Lys433, mediated by JNK, induces PKM2 detetramerization and promotes its nuclear translocation, a process essential for dendritic cell activation via modulation of glycolysis and fatty acid synthesis [53]. Similarly, p300-mediated acetylation at the same site disrupts FBP binding, stabilizes the low-activity dimeric form of PKM2, and enhances its nuclear localization, thereby facilitating HCC cell proliferation [54,55]. KAT8-catalyzed acetylation at Lys433 has also been linked to cisplatin resistance in lung cancer [56]. In contrast, Sirt6 removes the acetyl group from Lys433, promoting PKM2 export from the nucleus and suppressing its oncogenic activity [57]. Under high-glucose conditions, acetylation at Lys305 inhibits PKM2 enzymatic activity and accelerates its degradation via autophagy, ultimately enhancing the Warburg effect and tumor growth [58]. Additionally, Sirt1 deacetylates Lys135 and Lys206, thereby lowering PKM2 activity and suppressing glioma cell activation [59]. HDAC8-mediated deacetylation at Lys62 facilitates PKM2 nuclear translocation and upregulates oncogene transcription, contributing to HCC progression [20]. Overall, the dynamic acetylation and deacetylation of PKM2 play critical roles in modulating its structure, localization, and function, thereby influencing tumor development and offering promising avenues for targeted cancer therapies.

### 3.3. SUMOylation

SUMOylation of PKM2 is a critical post-translational modification that facilitates its nuclear translocation and modulates its function in tumor progression [60,61,62]. Specifically, SUMOylation at Lys270 promotes the structural shift of PKM2 from a tetrameric to a dimeric form, thereby enhancing its nuclear import. Replacement of wild-type PKM2 with a SUMOylation-deficient mutant (K270R) significantly reduces its regulatory effect on RUNX1 expression in leukemia cells [62]. Additionally, Lys336 has been identified as another SUMOylation site in A549 lung cancer cells. Modification at this residue redirects glucose metabolism from the tricarboxylic acid (TCA) cycle toward glycolysis and enhances HIF-1α transcriptional activity, further driving glycolytic flux and cancer progression [63].

### 3.4. Ubiquitination

Ubiquitination and deubiquitination are key regulatory mechanisms that modulate PKM2’s activity, stability, and subcellular localization—thereby influencing tumor progression and immune responses. Ubiquitination plays a pivotal role in controlling the dynamic balance between PKM2’s tetrameric and dimeric forms, thus shaping its metabolic functions in cancer cells [64,65,66]. DExD-box helicase 39B (DDX39B) directly interacts with PKM2 and protects it from STUB1-mediated ubiquitination and degradation, thereby stabilizing PKM2 protein levels. In addition, DDX39B promotes PKM2 nuclear translocation and activates glycolysis-related and oncogenic gene expression, independent of ERK1/2-mediated phosphorylation at Ser37 [67]. Similarly, the laforin/malin complex can induce PKM2 polyubiquitination, influencing its intracellular localization. On the other hand, several deubiquitinases enhance PKM2 stability and function. USP4 binds to PKM2 and catalyzes its deubiquitination, thereby upregulating glycolytic activity and reinforcing the Warburg effect, which contributes to tumor progression [68]. Other deubiquitinases—such as USP35, USP25, and OTUB2—also interact with PKM2, reduce its ubiquitination levels, increase enzymatic activity, and promote the pro-inflammatory activation of macrophages [69,70].

### 3.5. Oxidation

Oxidative modification, particularly at cysteine residues, plays a significant role in regulating PKM2 activity, thereby influencing cellular responses to oxidative stress and tumor progression [71]. PKM2 exists in three oxidative states—sulfation, disulfide, and polysulfide—primarily at three redox-sensitive cysteines: Cys358, Cys423, and Cys424. Among them, Cys424 is unique to PKM2 and undergoes sulfation, which increases the hydrophilicity of the dimer interface. This modification disrupts tetramer formation, reduces enzymatic activity, and enhances cellular resistance to oxidative stress [72]. Beyond tumor-related effects, oxidation of Cys423 and Cys424 during ischemic preconditioning enables PKM2 to function as a co-factor in reactive oxygen species (ROS) signaling. This interaction promotes the expression of HIF-1α-dependent genes, contributing to adaptive cellular responses under stress [73].

### 3.6. Other Modifications

In addition to the well-characterized modifications, other post-translational modifications—including succinylation, hydroxylation, and lactylation—also exert significant influence on PKM2 function [55,56,57]. Succinylation at lysine 498 enhances PKM2 enzymatic activity and promotes cancer cell proliferation [74]. Hydroxylation at proline residues 403 and 408 strengthens PKM2’s interaction with HIF-1α, thereby facilitating metabolic reprogramming [75]. Lactylation at lysine 62 modulates macrophage polarization and immune responses [76].

Collectively, these diverse PTMs form an intricate regulatory network that fine-tunes PKM2’s roles in cellular metabolism, tumor progression, and immune modulation. A deeper understanding of these modifications may reveal novel therapeutic opportunities, particularly for targeting PKM2 in liver-related diseases.

## 4. PKM2 Links Cellular Metabolism and Epigenetics

Epigenetics refers to heritable changes in gene expression and cellular phenotype that occur without alterations to the underlying DNA sequence. These changes—including DNA methylation, histone modifications, and chromatin remodeling—play essential roles in regulating gene transcription, maintaining cellular homeostasis, and contributing to disease pathogenesis [77].

These studies have revealed that PKM2, beyond its established functions in glycolysis, the tricarboxylic acid (TCA) cycle, and the pentose phosphate pathway, also acts at the intersection of metabolism and epigenetics. Through post-translational modifications, PKM2 can undergo conformational changes, translocate into the nucleus, and function as a transcriptional co-factor, thereby linking metabolic status to epigenetic regulation and gene expression (Figure 3).

### 4.1. Histone Modification Is the Bridge Linking PKM2 with Metabolism and Epigenetics

Histone and non-histone proteins undergo various PTMs, including acetylation, methylation, phosphorylation, and ubiquitination, which occur at specific amino acid residues. Among these, lysine acetylation and methylation in histones are the most commonly observed. These modifications alter chromatin structure and thereby regulate gene transcription and downstream signaling pathways. Traditionally, histone modifications are catalyzed by dedicated enzymes such as histone acetyltransferases (HATs) and HDACs. However, under certain conditions, non-canonical proteins with enzymatic activity can also influence histone modification levels. As previously discussed, nuclear translocation is essential for PKM2 to exert its non-metabolic functions. Notably, PKM2 acetylation enhances its nuclear accumulation without impairing its activity and even increases its protein kinase function. Once in the nucleus, PKM2 retains this kinase activity and directly regulates histone modifications, highlighting its multifaceted role at the interface of metabolism and epigenetic regulation.

Histone modifications within gene promoter regions play a critical role in regulating transcription. In glioma cells, it was first observed that PKM2, when phosphorylated by activated EGFR, translocates to the nucleus and interacts with histone H3, inducing H3 threonine 11 (T11) phosphorylation and promoting lysine 9 (K9) acetylation. This modification cascade enhances the transcription of cyclin D1 and c-Myc, thereby facilitating tumor cell proliferation, cell cycle progression, and glioma development [78]. Similar mechanisms have been identified in retinoblastomas and other cancers [79]. Moreover, EGFR-activated PKM2 can bind β-catenin, displacing HDAC3 from the CCND1 promoter. This displacement promotes histone H3 acetylation, increases cyclin D1 expression, and stimulates cell proliferation [80]. These findings suggest that nuclear PKM2 may similarly regulate histone modifications and gene transcription in liver diseases.

In HCC, PKM2-dependent phosphorylation of histone H3 at T11 has been shown to be essential for PD-L1 expression, establishing a mechanistic link between PKM2 activity and immune evasion [18]. EGF stimulation induces PKM2 nuclear translocation, which subsequently activates transcriptional programs. This supports the notion that PKM2-mediated gene regulation is responsive to growth signals [81]. Additionally, in HCC cells harboring the mutant p53 variant (N340Q/L344R), PKM2 expression and phosphorylation are elevated [82]. Unlike EGFR-mediated activation, mutant p53 promotes PKM2 tetramerization, which in turn drives H3-T11 phosphorylation, inhibits HDAC3 binding, stabilizes H3-K9 acetylation, and introduces K9 methylation—ultimately activating telomerase and promoting tumor growth. Similar regulatory patterns have been observed in liver fibrosis. High PKM2 expression in activated HSCs can be inhibited by TEPP-**46**, which prevents tetramer formation and modulates H3K9 acetylation. This results in reduced MYC and CCND1 expression, effectively suppressing HSC activation and fibrotic progression [80].

While current evidence highlights the role of PKM2 in modulating H3 acetylation and phosphorylation in liver diseases, further studies are needed to fully elucidate its epigenetic functions in other hepatic pathologies. Nevertheless, PKM2-mediated histone modification represents a promising therapeutic target in liver disease intervention.

### 4.2. PKM2 as a Co-Factor Regulates Non-Histone Epigenetics

In addition to modulating histone modifications and influencing gene expression, nuclear PKM2 also participates in the regulation of non-histone epigenetic mechanisms. For instance, alterations in DNA methylation at glycolysis-related promoter regions may disrupt the expression of key metabolic enzymes, thereby contributing to metabolic dysregulation and disease progression.

Activation of PKM2 modulates one-carbon metabolism by reducing methionine availability, which subsequently leads to hypomethylation of both nuclear and mitochondrial DNA. This epigenetic shift enhances mitochondrial biogenesis and function, and has been shown to boost CD8^+^ T cell recall responses and anti-tumor immunity, thereby improving the efficacy of adoptive cell therapies [83]. J The Jumonji C (JmjC) domain plays a key role in demethylase activation. JMJD5, a JmjC domain-containing dioxygenase, directly interacts with PKM2 to regulate cancer-associated metabolic reprogramming [84,85]. In addition, PKM2 facilitates the co-recruitment of HIF-1α and the transcriptional co-activator p300 to the hypoxia response element (HRE) of the PFKFB3 gene, inducing chromatin structural alterations and enhancing PFKFB3 transcription—thereby promoting the proliferation of hypoxic breast cancer cells [86]. Beyond recruitment, nuclear PKM2 in myocytes enhances the expression of subunits recognized by the SWI/SNF chromatin remodeling complex and facilitates their incorporation into regulatory regions of myogenic genes, ultimately promoting chromatin remodeling and transcriptional activation during muscle differentiation [87].

The interaction between nuclear PKM2 and HIF-1α enhances the binding of HIF-1α to the IL-1β promoter, thereby stimulating IL-1β transcription and secretion, promoting macrophage activation, and driving classical inflammatory responses [88]. Beyond its metabolic role, PKM2 also fine-tunes STAT3 activation, which is essential for Th17 cell differentiation and is implicated in autoimmune neuroinflammation [89,90]. Moreover, PKM2’s metabolic activity regulates ROS production in neutrophils, contributing to microbial clearance [91]. PKM2 also functions as a direct target of celastrol, a plant-derived triterpene. Covalent binding of celastrol at Cys31 induces conformational changes that inhibit PKM2 activity, subsequently reducing lipid accumulation, inflammation, and fibrosis in liver tissues [92]. I Recent studies have further demonstrated that PKM2 directly interacts with the histone methyltransferase EZH2 to mediate the epigenetic silencing of the carnitine transporter gene SLC6A9 [93]. Inhibition of PKM2 reverses this silencing effect, shifting tumor metabolism from aerobic glycolysis toward fatty acid β-oxidation. This metabolic reprogramming reveals novel therapeutic opportunities in tumor treatment.

## 5. Therapy Targeting PKM2

Therapeutic strategies targeting PKM2 can be classified into agonists and antagonists. Agonists promote the conversion of PKM2 from a dimer to a tetramer, thereby preventing its nuclear translocation. In contrast, inhibitors primarily prevent the formation of PKM2 tetramers; however, it remains unclear whether they influence the nuclear translocation of PKM2 (Table 2).

### 5.1. Agonists of PKM2

TEPP-**46** is a small molecule that activates and targets PKM2, capable of inducing PKM2 tetramer formation and enhancing its enzymatic activity by promoting PKM2 subunit interactions [102]. TEPP-**46** has been utilized to improve the progression of various diseases by activating PKM2 activity and promoting tetramer formation [88,103,104,105]. In liver diseases, TEPP-**46** has shown protective effects, including the inhibition of HCC proliferation by increasing PK activity and attenuating the Warburg effect [22,94]. Additionally, ex vivo studies have demonstrated that TEPP-**46** can prevent hepatic stellate cell activation and the development of hepatic fibrosis by inducing PKM2 tetramerization, thereby inhibiting the progression of hepatic fibrosis [34].

DASA-**58** is another well-established PKM2 activator that binds to the subunit interaction interface of PKM2, stabilizing its tetrameric form, enhancing PKM2 enzyme activity, and preventing the progression of transplanted malignancies in vivo [106]. Similarly to TEPP-**46**, DASA-**58** inhibits HCC proliferation and mitigates the progression of hepatic inflammatory diseases by maintaining PKM2 in its tetrameric state [107,108]. Additionally, in hepatic macrophages, DASA-**58** promotes enzyme activity, which inhibits inflammatory macrophage differentiation and alleviates hepatic fibrosis [31].

ML**265** binds to the PKM2 dimer-dimer interface, promoting the formation of tetrameric PKM2 and influencing cellular function [109]. In the CCL4-induced liver regeneration model, ML**265** treatment increased the nuclear distribution of PKM2, which inhibited liver regeneration [96].

TP-1454 is a potent PKM2 activator that enhances the binding and activation of two dimeric PKM2 and entered clinical trials as the first oral PKM2 activator for the treatment of advanced solid tumors [110]. In addition, Mitapivat (AG-348), a novel oral pyruvate kinase variant activator, has been shown in multiple clinical trials to play a role in diseases such as pyruvate kinase deficiency [81,111,112].

Various amino acids can also bind to PKM2 to activate it. For instance, serine, aspartic acid, and asparagine can undergo conformational changes to activate PKM2, thereby enhancing its ability to bind to substrates [113].

### 5.2. Inhibitors of PKM2

Shikonin and compound 3k are considered effective PKM2 inhibitors that induce apoptosis in HCC cells in vitro and enhance the antitumor effects of sorafenib in vivo [21,97,100,114,115]. In vivo and in vitro experiments based on mouse colon cancer showed that shikonin disrupts PKM2 dimer and tetramer formation and induces the formation of macromolecular complexes, thereby inhibiting PKM2 activity and glycolytic flux and ameliorating colitis progression [116]. To improve the effect of shikonin, researchers designed MPDA nanoparticles loaded with shikonin and hyaluronic acid modification for the treatment of colorectal cancer liver metastasis by inhibiting PKM2 for immunometabolic reprogramming [117]. Compound **3**k was more selective for PKM2 than shikonin, and played a disease-protecting role by promoting the disruption of PKM2 tetramer [100,104].

Based on the chemical structure similar to that of alizarin, it has been found that vitamin K, especially vitamins K3 and K5, significantly inhibit PKM2, but have little effect on PKM1 and PKL, and are able to act as co-factors in tumor therapy [99]. Additionally, alkannin, a natural compound structurally similar to shikonin, acts as a tumor-specific inhibitor of PKM2. It significantly reduces the glycolytic rate, as manifested by cellular lactate production and glucose consumption in drug-sensitive and resistant cancer cell lines that predominantly express PKM2 [101]. This highlights its potential for future clinical applications.

### 5.3. Regulator of PKM2

In addition to classical activators and inhibitors, membrane-bound protein A5 regulates hepatic macrophage polarization by directly blocking PKM2 Y105 phosphorylation and nuclear translocation, thereby attenuating HFD-induced NASH [39].Simvastatin sensitizes HCC cells to sorafenib by inhibiting HIF1-α/PPAR-γ/PKM2-mediated glycolysis. Simvastatin can sensitize HCC cells to sorafenib by inhibiting HIF1-α/PPAR-γ/PKM2-mediated glycolysis [27]. Plant-derived triterpene celastrol can covalently bind to PKM2 residue Cys 31, alter the spatial binding of PKM2, inhibit its PK activity, and improve NALFD [92].

Metformin, currently utilized in the management of type 2 diabetes, has been extensively investigated as a metabolic modulator. Studies have demonstrated that metformin significantly downregulates the expression of PKM2, thereby sensitizing tumor cells to chemotherapeutic agents and exhibiting potential anti-cancer effects [118,119]. Epidemiological evidence further supports its promising therapeutic potential as an anti-cancer agent [120].

Flavonoid compounds like quercetin have been shown to inhibit PKM2 activity, thereby affecting tumor cell proliferation and migration [121]. Curcumin, derived from turmeric (Curcuma longa), regulates PKM2 activity to suppress glycolysis, exhibiting anti-cancer properties [122]. Furthermore, extracts from Lycium barbarum also inhibit PKM2, further alleviating the inflammatory response [123]. These herbal remedies offer promising new strategies for cancer treatment by modulating PKM2 activity, with significant potential for clinical application.

In addition to its function by affecting PKM2 enzyme activity, digoxin is able to bind to PKM2, preventing its binding to histones on the basis of not affecting its enzyme activity, which in turn inhibits the nuclear translocation of PKM2 to HIF-1α and attenuates macrophage glycolysis and inflammatory differentiation [37]. Cynaroside, a flavonoid compound, promoted macrophage phenotypic transition from pro-inflammatory M1 to anti-inflammatory M2, and mitigated sepsis-associated liver inflammatory damage [124].

In conclusion, agonists and antagonists targeting PKM2 can play a therapeutic role in tumors and liver diseases. However, clinical trials for PKM2 therapy are still insufficient, and further research on the occurrence of signaling pathway cascade induced by PKM2 therapy is also needed. Therefore, in-depth exploration of the mechanism of PKM2 in disease progression through combined adjuvant therapy may be a more effective therapeutic measure.

## 6. Risks and Challenges

PKM2 has attracted considerable interest in hepatology owing to its dual capacity as a metabolic enzyme and a transcriptional regulator. In metabolic liver disease, PKM2 modulates glycolytic flux and redox equilibrium, thereby influencing ATP generation and ROS accumulation, both of which are pivotal determinants of hepatocyte viability [125]. In liver fibrosis, PKM2 contributes to the activation and metabolic reprogramming of hepatic stellate cells, fostering extracellular matrix deposition and fibrogenesis [34]. In HCC, PKM2 sustains the “Warburg effect” while also translocating to the nucleus, where it cooperates with transcription factors such as HIF-1α to drive oncogenic transcriptional programs, ultimately promoting proliferation, angiogenesis, immune evasion, and therapeutic resistance [126]. Taken together, these findings highlight the potential of PKM2 as both a biomarker and a therapeutic target across diverse liver pathologies.

Despite its apparent significance, PKM2 remains a subject of debate. Evidence from several cancer models indicates that PKM2 may be dispensable or functionally redundant, with PKM1 or alternative metabolic pathways compensating for its loss [127]. This redundancy raises the question of whether PKM2 inhibition alone can yield meaningful therapeutic benefits in liver disease. Moreover, PKM2 function appears highly context- and tissue-dependent, exerting protective effects under certain stress conditions yet pathogenic roles in others, which complicates its translational potential [128]. Another major challenge lies in the specificity of pharmacological targeting, as PKM2 shares substantial structural homology with other isoforms, thereby complicating the development of selective modulators and raising concerns about off-target effects.

Future investigations should aim to delineate the precise contexts in which PKM2 exerts indispensable functions in the liver. Approaches such as multi-omics profiling, conditional knockout models, and single-cell resolution analyses may provide critical insights into its cell type-specific and stage-dependent roles. Such studies will be essential for determining whether PKM2 constitutes a universal therapeutic target or whether its clinical relevance is confined to particular subsets of liver diseases.

## 7. Concluding Remarks

The incidence rate of most liver diseases in the population is gradually increasing, and there is a lack of effective or approved treatment methods. Glycolysis plays an important role in the progression of liver diseases, especially HCC, promoting the body’s inflammatory response and the growth, metastasis, and drug resistance of tumor cells. PKM2, one of the rate-limiting enzymes of glycolysis, has a unique structure and function. PKM2 plays different functional roles in cells based on its different aggregation states. PKM2 in tetramer form has high PK activity, promoting the transfer of pyruvate to the mitochondria to participate in the tricarboxylic acid cycle and promoting cellular oxidative phosphorylation. Conversely, PKM2 in dimer form promotes cellular glycolysis, leading to the accumulation of products from glycolysis, which is conducive to the synthesis of macromolecules and promotes cell proliferation. Not only that, dimer PKM2 can be transported to the nucleus of the cell, mediating the transcription of downstream oncogenes, cell proliferation genes, glycolysis genes, and inflammation genes, guiding disease progression. Here, we deeply explore the post-translational modifications on PKM2 structure, location and activity, and discuss its role as a link between metabolism and epigenetics in liver disease and other diseases. Given the important role of PKM2, targeting PKM2 may be a potential approach to treating liver disease. However, PKM2 has numerous modification sites, so how to effectively regulate PKM2 is an important issue to be resolved in the future. In addition, how PKM2’s multiple modification sites interact with conformational modulators is a new direction for developing effective PKM2 conformational modulators.

In summary, although modulators targeting PKM2 still require extensive research, PKM2 could be a potentially important target for liver disease.

## Figures and Tables

**Figure 1 biomolecules-15-01327-f001:**
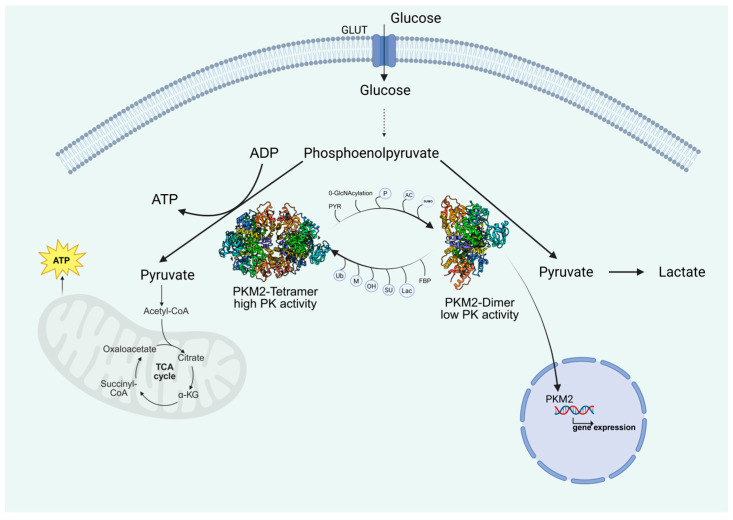
Role of PKM2 in the glycolysis pathway and relationship between PKM2 enzyme activity and spatial conformation. Phosphoenolpyruvate is converted to pyruvate by the highly active PKM2 tetramer, which is then transported to mitochondria to take part in the tricarboxylic acid cycle (TCA) and provide energy. However, the low-active PKM2 dimer leads to the accumulation of upstream products of glycolysis and provides substrates for biosynthesis, while the dimeric form of PKM2 will undergo nuclear translocation and promote the transcription. FBP, fructose 1,6-bisphosphate; PYR, pyruvate.

**Figure 2 biomolecules-15-01327-f002:**
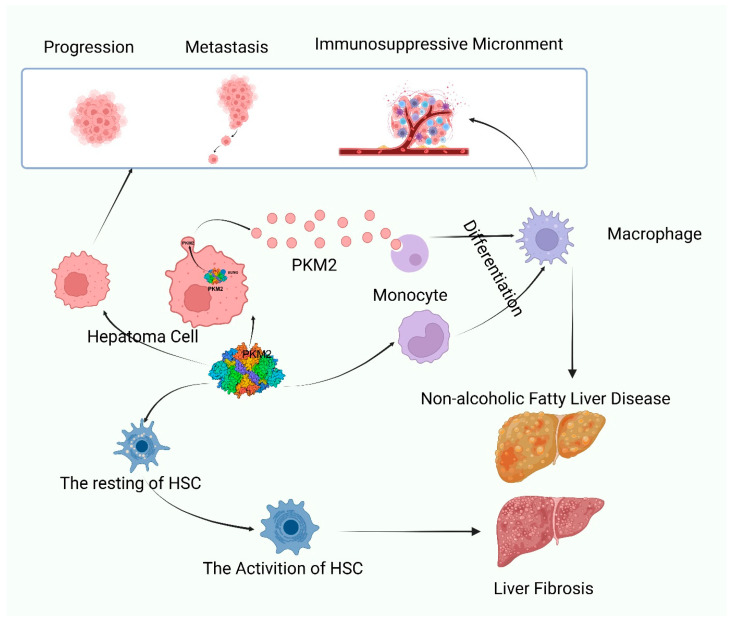
Role of PKM2 in liver diseases. PKM2 is involved in the process of cell proliferation, differentiation and activation in hepatocellular carcinoma, hepatic macrophages, hepatic stellate cells and other cells involved in the progression of liver diseases. HSC, hepatic stellate cell.

**Figure 3 biomolecules-15-01327-f003:**
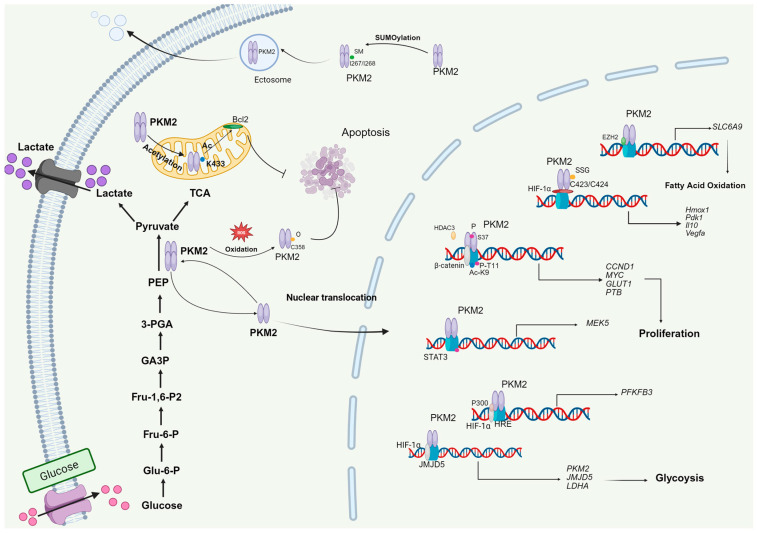
Metabolic and non-metabolic roles of PKM2. After glucose enters the cell through the GLUT1 receptor, a series of enzymatic reactions generate ATP and various metabolites. PKM2 catalyzes the production of pyruvate. Through various post-translational modifications, PKM2 can transform from a tetramer to a dimer, enter the cell nucleus, promote gene synthesis, and exert non-metabolic functions. Furthermore, acetylated or oxidized PKM2 can promote apoptosis through various pathways.

**Table 1 biomolecules-15-01327-t001:** Post-translational modifications of PKM2.

Modification	Specific Site	Effects
Phosphorylation	Tyr105 Ser37	Promotes the formation of low-activity PKM2 dimers
		Promotes dimer formation
		Promotes PKM2 nuclear localization to exert non-metabolic effects
	Thr365	Enhances enzymatic activity
Acetylation	Lys433	Inhibits enzymatic activity
		promotes dimer nuclear translocation
		Mitochondrial PKM2 inhibits apoptosis
	Lys305	Reduces enzymatic activity; increases lysosomal degradation
	Lys66	Increases PKM2 expression
Ubiquitination	Lys186 Lys206	Regulates PKM2 activity; induces tetramer formation
SUMOylation	Lys270	Induces dimer formation and nuclear translocation
	Lys336	Promotes glycolysis; enhances HIF-1α transcriptional activity
O-GlcNAcylation	Ser362 Thr365	Blocks tetramer formation; decreases enzymatic activity
Oxidation	Csy424	Reduces tetramer formation; confers resistance to oxidative stress
		Reacts with transient ROS to promote HIF-1α expression
	Met239	Promotes tetramer formation; increases enzymatic activity
	Csy358	Responds to ROS and inhibits apoptosis
Lactylation	Lys62	Promotes tetramer formation and enzymatic activity; inhibits nuclear translocation
Hydroxylation	Pro403/408	Enhances interaction with HIF-1α and promotes metabolic reprogramming
	Lys66	Induces lysosomal degradation of PKM2

**Table 2 biomolecules-15-01327-t002:** The main activators and inhibitors of PKM2.

Compound	Effects	Disease
Activators		
TEPP-**46** 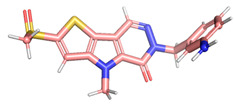	Promoting PKM2 tetramer formation and increasing PKM2 enzymatic activity	HCC [94] NAFLD [95] Liver Fibrosis [34]
DASA-**58** 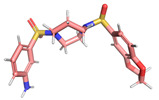	Promoting PKM2 tetramerization and reducing lactate scretion	Liver Fibrosis [31] HCC [94]
ML**265** 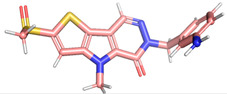	Binding at the PKM2 dimer-dimer interface induces the formation of tetrameric PKM2	NASH [96]
Inhibitors		
Shikonin 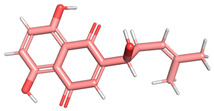	Decreasing PKM2 enzymatic activity	HCC [97] NAFLD [98] Liver Fibrosis [34]
Vitamin K**3** 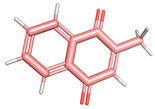	Decreasing PKM2 enzymatic activity	HCC [99]
Compound **3**k 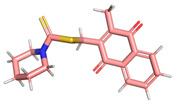	Induction of PKM2 tetramer disruption	HCC [100]
Digoxin 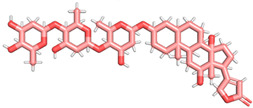	Preventing PKM2 from binding to histones and thus inhibiting its nuclear translocation to HIF-1α	NASH [37]
Alkannin 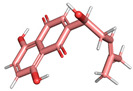	Decreasing PKM2 enzymatic activity	HCC [101]

To facilitate clarity, activators and inhibitors are presented separately and use gray to emphasize.

## Data Availability

No new data were created or analyzed in this study. Data sharing is not applicable to this article.

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
