# Peer review of "Pyruvate Kinase M2 Links Metabolism and Epigenetics: A New Target for Liver Disease Treatment"

_biomolecules, 2025, doi:10.3390/biom15091327_

Round 1
Reviewer 1 Report
Comments and Suggestions for Authors
This is a comprehensive and timely review of PKM2’s roles in liver diseases. Manuscript is well-written by the authors with mechanistic insights. However, there are a few areas where I believe that authors can work on the content, and could be expanded for better clarity.
There are cases where PKM2 is non-essential or redundant in certain cancer models, but I have no idea in liver diseases. Therefore, in my opinion it will better if authors can discuss on PKM2 controversies and probable challenges. Discussing this in the review will balance the context and strengthen the PKM2’s significance.
I think the part on PKM2 as a biomarker needs to be elaborated. Yiu can mention some clinical studies here.
Post-translational modifications part can be summarized into one additional table or schematic, which could greatly help readers. Because this section is quite dense for reading.
I also suggest authors to design a new figure illustrating how PKM2 bridges metabolism and epigenetic regulation, or you can alter the figure 3 with this concept. Please make sure if you edit the figure 3, it shouldn’t be too complicated to understand.
Figure 3 legend is very difficult to understand. It needs to be revised. You can consider breaking into two sentences.
In table 1 “lactate scretion” should be “lactate secretion”.
“NALFD” should be “NAFLD”.
“HIF1-α” should be HIF-1α
You have mentioned ChatGPT-4 in Acknowledgments: In the preparation of this manuscript, the authors used ChatGPT-4 for manuscript polishing. The authors have reviewed and edited the output and take full responsibility for the content of this publication.
Though you have appropriately acknowledged the ChatGPT, but I don’t think this is ethically right. However, I will leave a comment for editor to decide regarding the acknowledgement statement.
Author Response
Comment 1: There are cases where PKM2 is non-essential or redundant in certain cancer models, but I have no idea in liver diseases. Therefore, in my opinion it will better if authors can discuss on PKM2 controversies and probable challenges. Discussing this in the review will balance the context and strengthen the PKM2’s significance.
Response: Thank you for this valuable suggestion. To strengthen the discussion on the role of PKM2 and to address its related controversies and potential challenges, we have added a new Section 6 entitled “Risks and Challenges.” In this section, we elaborate on the context-dependent roles of PKM2, its possible redundancy with other isoforms, and the difficulties associated with its therapeutic targeting. This addition aims to provide a more balanced perspective and better illustrate the potential controversies and challenges surrounding PKM2.
Comment 2: The part on PKM2 as a biomarker needs to be elaborated. You can mention some clinical studies here
Response: We appreciate the reviewer’s constructive suggestion. In response, we have expanded Section 2 by incorporating relevant clinical studies into the disease description. These additions further strengthen the discussion of PKM2 as a potential biomarker in liver diseases, providing more comprehensive clinical evidence to support its diagnostic and prognostic value.
Comment 3: Post-translational modifications part can be summarized into one additional table or schematic, which could greatly help readers. Because this section is quite dense for reading
Response: Thank you for the helpful suggestion. In the revised manuscript, we have replaced the original schematic diagram with a table to facilitate clearer presentation and improve reader comprehension.
Comment 4: I also suggest authors to design a new figure illustrating how PKM2 bridges metabolism and epigenetic regulation, or you can alter the figure 3 with this concept. Please make sure if you edit the figure 3, it shouldn’t be too complicated to understand.
Response: Thank you for the insightful comment. In the revised manuscript, we have designed a new figure to illustrate how PKM2 links metabolic regulation with epigenetic control. This addition aims to provide a clearer conceptual framework and enhance the readers’ understanding of PKM2’s integrative role.
Comment 5: In table 1 “lactate scretion” should be “lactate secretion”.
“NALFD” should be “NAFLD”.
“HIF1-α” should be HIF-1α
Response: We appreciate the reviewer’s careful observation. The academic terminology errors in the table have been corrected in the revised manuscript to ensure accuracy and consistency.
Comment 6: You have mentioned ChatGPT-4 in Acknowledgments: In the preparation of this manuscript, the authors used ChatGPT-4 for manuscript polishing. The authors have reviewed and edited the output and take full responsibility for the content of this publication.
Response: Thank you for your valuable suggestions. We used ChatGPT to polish the language to ensure clear and accurate expression without affecting the essence of the research; all the substantive content still belongs to our own creative achievements. We have attached the relevant instructions on the use of ChatGPT as requested by the editors and the magazine.
Reviewer 2 Report
Comments and Suggestions for Authors
In this review article, the authors focus on the role of PKM2 in liver diseases, including hepatocellular carcinoma, and discuss its significance as a therapeutic target for these diseases. The information is well organized based on many findings, making this a useful review for understanding both pathological mechanisms and clinical applications. Minor improvements are suggested below.
Many excellent reviews on pyruvate kinase M2 and hepatocellular carcinoma/liver disease have already been published (Jadhav M, et al. Gene Expr. 2023; Qu H, et al. Cells. 2023; Wan Q, et al. Tissue Eng Part C Methods. 2025, etc.). What unique perspectives does your review offer compared to those reviews? The authors should state the novel perspectives of your review in the introduction to differentiate it from previous reviews. The term "hepatocellular carcinoma (HCC)" appears frequently in this manuscript. Once a word has been abbreviated, it must be written in its abbreviation from then on. The quality of the structural formula images in Table 1 is not uniform. For example, DASA-58 is less clear than ML265, while Shikonin is larger and clearer. All structural formulas should be drawn using the same software as the original.
Author Response
Comment 1: Many excellent reviews on pyruvate kinase M2 and hepatocellular carcinoma/liver disease have already been published (Jadhav M, et al. Gene Expr. 2023; Qu H, et al. Cells. 2023; Wan Q, et al. Tissue Eng Part C Methods. 2025, etc.). What unique perspectives does your review offer compared to those reviews? The authors should state the novel perspectives of your review in the introduction to differentiate it from previous reviews. The term "hepatocellular carcinoma (HCC)" appears frequently in this manuscript.
Response: We thank the reviewer for this important comment. We fully acknowledge the existence of several comprehensive reviews on PKM2 in liver diseases. To emphasize the novelty of our review, we have revised the certain sections to clearly highlight our unique perspectives:
- Integration of metabolism and epigenetics: While previous reviews primarily focused on PKM2 as a metabolic regulator in HCC or other liver diseases, our review specifically emphasizes the dual role of PKM2 in bridging cellular metabolism with epigenetic regulation. We summarize evidence showing how PKM2-mediated metabolic intermediates and nuclear functions influence histone modification, DNA methylation, and chromatin remodeling in the liver.
- Beyond hepatocellular carcinoma: In addition to HCC, our review extends the discussion to other liver pathologies, including liver fibrosis, acute liver failure and metabolic liver diseases, areas that have been less systematically addressed in previous reviews.
- Therapeutic implications: We provide a forward-looking discussion on PKM2 as a potential therapeutic target, not only by inhibiting its metabolic functions but also by modulating its epigenetic regulatory roles.
These revisions distinguish our review by framing PKM2 as a metabolic–epigenetic nexus with broad implications for liver disease pathogenesis and therapy, thereby complementing and extending prior reviews.
Comment 2: Once a word has been abbreviated, it must be written in its abbreviation from then on. The quality of the structural formula images in Table 1 is not uniform. For example, DASA-58 is less clear than ML265, while Shikonin is larger and clearer. All structural formulas should be drawn using the same software as the original.
Response: We thank the reviewer for this helpful comment. We have carefully checked and corrected all abbreviations throughout the manuscript to ensure consistency. In addition, the structural formulas in Table 1 have been replaced with higher-quality 3D structures to improve clarity and visualization.
Round 2
Reviewer 1 Report
Comments and Suggestions for Authors
Manuscript is significantly improved by the authors. However, while reading the newly added section 6, I could not see a single reference in the whole section. There are many scientific statements in the whole section which requires references. Please recheck the section and cite the relevant work appropriately. Now the manuscript requires minor revision. Please revise the section 6 for references.
Author Response
Manuscript is significantly improved by the authors. However, while reading the newly added section 6, I could not see a single reference in the whole section. There are many scientific statements in the whole section which requires references. Please recheck the section and cite the relevant work appropriately. Now the manuscript requires minor revision. Please revise the section 6 for references.
Response: Thank you for your valuable comments. We have revised Section 6 according to your suggestions. Specifically, we have improved the content and inserted the relevant references to support the discussion.